# Deep Implicit Optimization for Robust and Flexible Image Registration

## Abstract

Deep Learning in Image Registration (DLIR) methods have been tremendously successful in image registration due to their speed and ability to incorporate weak label supervision at training time. However, DLIR methods forego many of the benefits of classical optimization-based methods. The functional nature of deep networks do not guarantee that the predicted transformation is a local minima of the registration objective, the representation of the transformation (displacement/velocity field/affine) is fixed, and the networks are not robust to domain shift. Our method aims to bridge this gap between classical and learning methods by incorporating optimization as a layer in a deep network. A deep network is trained to predict multi-scale dense feature images that are registered using a black box iterative optimization solver. This optimal warp is then used to minimize image and label alignment errors. By *implicitly* differentiating end-to-end through an iterative optimization solver, our learned features are registration and label-aware, and the warp functions are guaranteed to be local minima of the registration objective in the feature space. Our framework shows excellent performance on in-domain datasets, and is agnostic to domain shift such as anisotropy and varying intensity profiles. For the first time, our method allows switching between arbitrary transformation representations (free-form to diffeomorphic) at test time with zero retraining. End-to-end feature learning also facilitates interpretability of features, and out-of-the-box promptability using additional label-fidelity terms at inference.

## 1 Introduction

Deformable Image Registration (DIR) refers to the local, non-linear alignment of images by estimating a dense displacement field. Many workflows in medical image analysis require images to be in a common coordinate system for comparison, analysis, and visualization, including comparing inter-subject data in neuroimaging [53, 104, 97, 38, 89, 94], biomechanics and dynamics of anatomical structures including myocardial motions, airflow and pulmonary function in lung imaging, organ motion tracking in radiation therapy [78, 77, 11, 70, 29, 105, 50, 18, 71, 84], and life sciences research [112, 104, 99, 80, 98, 72, 17].

Classical DIR methods are based on solving a variational optimization problem, where a similarity metric is optimized to find the best transformation that aligns the images. However, these methods are typically slow, and cannot leverage learning to incorporate a training set containing weak supervision such as anatomical landmarks or expert annotations. The quality of the registration is therefore limited by the fidelity of the intensity image. Deep Learning for Image Registration (DLIR) is an interesting paradigm to overcome these challenges. DLIR methods take a pair of images as input to a neural network and output a warp field that aligns the images, and their associated anatomical landmarks. The neural network parameters are trained to minimize the alignment loss over image pairs and landmarks in a training set. A benefit of this method is the ability to incorporate weak

supervision like anatomical landmarks or expert annotations during training, which performs better landmark alignment without access to landmarks at inference time.

**Motivation.** However, DLIR methods face several limitations. First, the prediction paradigm of deep learning implies the feature learning and amortized optimization steps are fused; transformations predicted at test-time may not even be a local minima of the alignment loss between the fixed and moving image. The end-to-end prediction also implies that the representation of the transformation is fixed (as a design choice of the network), and the model cannot switch between different representations like free-form, stationary velocity, geodesic, LDDMM, B-Splines, or affine at test time without additional finetuning, in sharp contrast to the flexibility of classical methods. Typical registration workflows require a practitioner to try different parameterizations of the transformation (free-form, stationary velocity, geodesic, LDDMM, B-Splines, affine) to determine the representation most suitable for their application and additional retraining becomes expensive. Moreover, design decisions like sparse keypoint learning for affine registration [103, 16, 69, 40] do not facilitate dense deformable registration. Furthermore, DLIR methods do not allow interactive registration using additional landmarks or label maps at test time, which is crucial for clinical applications. Hyper-parameter tuning for regularization is also expensive for DLIR methods. Although recent methods propose conditional registration [44, 67] to amortize over the hyperparameter search during training, the family of regularization is fixed in such cases, and space of hyperparameters becomes exponential in the number of hyperparameter families considered. Lastly, current DLIR methods are not robust to minor domain shifts like varying anisotropy and voxel resolutions, different image acquisition and preprocessing protocols [62, 53, 70, 43]. Robustness to domain shift is imperative to biomedical and clinical imaging where volumes are acquired with different scanners, protocols, and resolutions, where the applicability of DLIR methods is limited to the training domain.

**Contributions.** We introduce *DIO*, a generic *differentiable implicit optimization* layer to a learnable feature network for image registration. By decoupling feature learning and optimization, our framework **incorporates weak supervision like anatomical landmarks into the learned features** during training, which improves the fidelity of the feature images for registration. Feature learning also leads to *dense* feature images, which smoothens the optimization landscape compared to intensity-based registration due to homogenity present in most medical imaging modalities. Since optimization frameworks are agnostic to spatial resolutions and feature distortions, DIO is extremely robust to domain shifts like varying anisotropy, difference in sizes of fixed and moving images, and different image acquisition and preprocessing protocols, even when compared to models trained on contrast-agnostic synthetic data [43]. Moreover, our framework allows *zero-cost plug-and-play* of arbitrary transformation representations (free-form, geodesics, B-Spline, affine, etc.) and regularization at test time without additional training and loss of accuracy. This also paves the way for practitioners to perform **quick and interactive registration**, and use **additional arbitrary 'prompts'** such as new landmarks or label maps out-of-the-box at test time, as part of the optimization layer.

## 2 Related Work

**Deep Learning for Image Registration** DIR refers to the alignment of a fixed image $I_f$ with a moving image $I_m$ using a transformation $\varphi \in T$ where $T$ is a family of transformations. Classical methods formulate a variational optimization problem to find the optimal $\varphi$ that aligns the images [15, 4, 7, 5, 6, 2, 15, 25, 24, 23, 27, 39, 63, 102, 101, 100, 46, 60, 61, 76, 33, 32, 12]. In contrast, earliest DLIR methods used supervised learning [19, 55, 82, 88] to predict the transformation $\varphi$. Voxelmorph [13] was the first unsupervised method utilizing a UNet [83] for unsupervised registration on brain MRI data. Recent works considered different architectural designs [21, 56, 48, 66], cascade-based architectures and loss functions [116, 115, 49, 26, 68, 114, 79, 20], and symmetric or inverse consistency-based formulations [65, 51, 52, 92, 116]. [67, 44] inject the hyperparameter as input and perform amortized optimization over different values of the hyperparameter. Domain randomization and finetuning [43, 96, 73, 30] are also proposed to improve robustness of registration to domain shift, that is a core necessity in medical imaging since different institutions follow varying acquisition and preprocessing pipelines. Foundational models are also proposed to improve registration accuracy [57, 93]. Another line of work propose to use the implicit priors of deep learning [95] within an optimization framework [110, 106, 49, 45]. We refer the reader to [36, 41, 28] for other detailed reviews.

**Iterative methods for DLIR** Owing to the success of iterative optimization methods, few DLIR methods propose emulating the iterative optimization within a network. [115, 116] use a cascade of

networks to iteratively predict a warp field, and use the warped moving image as the input to the next layer in the cascade. TransMorph-TVF [20] uses a recurrent network to predict a time-dependent velocity field. [114] use a shared weights encoder to output feature images at multiple scales, and a deformation field estimator utilizing a correlation layer. RAFT [91] similarly builds a 4D correlation volume from two 2D feature maps, and updates the optical flow field using a recurrent unit that performs lookup on the correlation volume. However, such recursive formulations have a large memory footprint due to explicit backpropagation through the entire cascade [8], and are not adaptive or optimal with respect to the inputs. In contrast, DIO uses optimization as a layer – guaranteeing convergence to a local minima, and *implicit differentiation* avoids storing the entire computation graph making the framework both memory and time efficient.

**Feature Learning for Image Registration** [103, 16, 69, 40] learn keypoints from images which is then used to compute the optimal affine transform using a closed form solution. However, these methods are restricted to transformations that can be represented by differentiable *closed-form* analytical solutions, making backpropagation trivial. These sparse keypoints cannot be reused for dense deformable registration either. On the other hand, dense deformable registration (diffeomorphic or otherwise) is almost universally solved using iterative optimization methods. This motivates the need to perform *implicit differentiation* through an iterative optimization solver to perform feature learning for registration. Other approaches learn image features to perform registration [108, 59, 107, 81], but do not perform feature learning and registration end-to-end, i.e., the features obtained are not task-aware and may not be optimal for registration, especially for anatomical landmarks. Learned features are either fed into a functional form to compute the transformation end-to-end, or are learned using unsupervised learning in a stagewise manner. In contrast, by implicitly differentiating through a black-box iterative solver, and minimizing the image and label alignment losses end-to-end, DIO learns features that are *registration-aware*, *label-aware*, and *dense*. The optimization routine also guarantees that the transformation is a local minima of the alignment of high-fidelity feature images.

**Deep Equilibrium models** Deep Equilibrium (DEQ) models [9, 34] have emerged as an interesting alterative to recurrent architectures. DEQ layers solve a fixed-point equation of a layer to find its equilibrium state without unrolling the entire computation graph. This leads to high expressiveness without the need for memory-intensive backpropagation through time [10, 8, 31, 75, 37, 111]. PIRATE [45] uses DEQ to finetune the PnP denoiser network for registration, but unlike our work, the data-fidelity term comes from the intensity images. However, these methods use DEQ to emulate an infinite-layer network, which typically consists of learnable parameters within the recurrent layer. **Conceptually, our work does not aim to simply emulate such an infinite cascade, but rather use DEQ to *decouple* feature learning and optimization in an end-to-end registration framework.** This inherits all the robustness and agnosticity of optimization-based methods, while retaining the fidelity of learned features. DEQ allows us to avoid the layer-stacking paradigm for cascades, and use optimization as a black box layer without storing the entire computation graph, leading to constant memory footprint and faster convergence. This allows learnable features to be registration-aware since gradients are backpropagated to the feature images through the optimization itself.

## 3 Methods

The registration problem is formulated as a variational optimization problem:

$$\varphi^* = \arg\min_{\varphi} L(I_f, I_m \circ \varphi) + R(\varphi) = \arg\min_{\varphi} C(\varphi, I_f, I_m) \tag{1}$$

where $I_f$ and $I_m$ are fixed and moving images respectively, $L$ is a loss function that measures the dissimilarity between the fixed image and the transformed moving image, and $R$ is a suitable regularizer that enforces desirable properties of the transformation $\varphi$. We call this the *image matching* objective. If the images $I_f$ and $I_m$ are supplemented with anatomical label maps $L_f$ and $L_m$, we call this the *label matching* objective. Classical methods perform image matching on the intensity images, but the label matching performance is bottlenecked by the fidelity of image gradients with respect to the label matching objective, and dynamics of the optimization algorithm. Deep learning methods mitigate this by injecting label matching objectives (for example, Dice score) into the objective Eq. (1) and using a deep network with parameters $\theta$ to predict $\varphi$ for every image pair as input. In essence, learning-based problems solve the following objective:

$$\theta^* = \arg\min_{\theta} \sum_{f,m} L(I_f, I_m \circ \varphi_\theta) + D(S_f, S_m \circ \varphi_\theta) + R(\varphi_\theta) = \arg\min_{\theta} \sum_{f,m} T(\varphi_\theta, I_f, I_m, S_f, S_m) \tag{2}$$

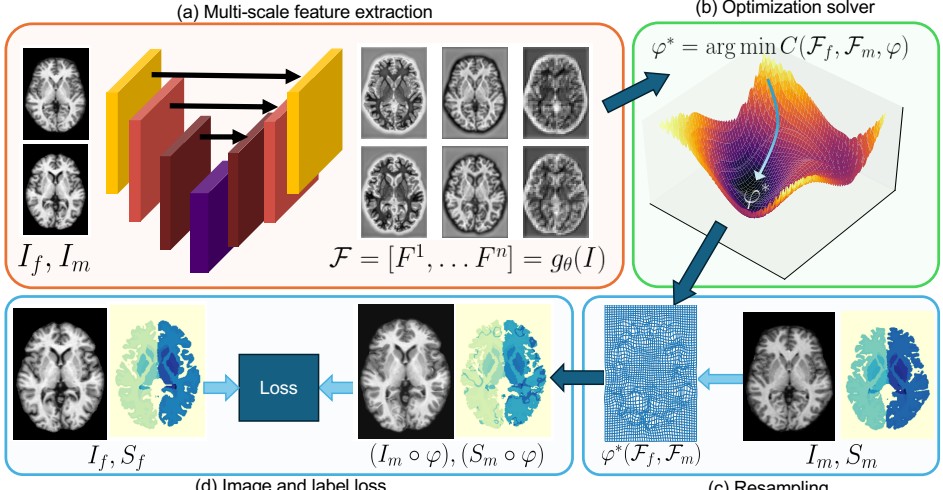

Figure 1: **Overview of our framework**. **(a)** A neural network extracts multi-scale features from the input images. **(b)**These features are used to optimize warp fields using a multi-scale differentiable optimization solver. **(c)** The optimized transform is used to warp the moving image and labels. **(d)** The warped image/label are compared with the fixed image/label using a similarity metric.

where $\varphi_\theta(I_f, I_m)$ is abbreviated to $\varphi_\theta$. This leads to learned transformations $\varphi_\theta$ that perform both good image and label matching. However, the feature learning and optimization are coupled, and the learned features are optimized only for a specific training domain. This limitation primarily marks the difference between DIO and existing DLIR methods.

Fig. 1 shows the overview of our method. Our goal is to learn feature images such that **registration in this feature space corresponds to both image and label matching performance**, by disentangling feature learning and optimization. We do this by using a feature network to extract dense features from the intensity image, that are used to solve Eq. (1) using a black-box optimization solver, and obtain an optimal transform $\varphi^*$. Once $\varphi^*$ is obtained, this is plugged into Eq. (2) to obtain gradients with respect to $\varphi^*$. Since $\varphi^*$ is a function of the feature images, we *implicitly differentiate* through the optimization to backpropagate gradients to the feature images and to the deep network. We discuss the details of our method in the following sections.

### 3.1 Feature Extractor Network

The first component of our framework is a feature network that extracts dense features from the intensity images. This network is parameterized by $\theta$, and takes an image $I \in \mathbb{R}^{H \times W \times D \times C_{in}}$ as input and outputs a feature map $F \in \mathbb{R}^{H \times W \times D \times C}$, where $C$ is the number of feature channels, i.e. $F = g_\theta(I)$. Unlike existing DLIR methods where moving and fixed images are concatenated and passed to the network, our feature network processes the images *independently*. This allows the fixed and moving images to be of different voxel sizes. The feature network can also output multi-feature feature maps $\mathcal{F} = g_\theta(I) = [F^0, F^1, \ldots, F^N]$, where $F^k \in \mathbb{R}^{H/2^k \times W/2^k \times D/2^k \times C_k}$, which can be used by multi-scale optimization solvers. The feature network is agnostic to architecture choice, and we ablate on different architectures in the experiments.

### 3.2 Implicit Differentiation through Optimization

Given the feature maps $F_f$ and $F_m$ extracted from the fixed and moving images, an optimization solver optimizes Eq. (1) to obtain the transformation $\varphi^*$. This can be written by modifying Eq. (1) to use the feature maps $F$; i.e. $\varphi^* = \arg\min_\varphi C(F_f, F_m \circ \varphi)$. A local minima of this equation satisfies:

$$\varrho(\varphi^*, F_f, F_m) = \left.\frac{\partial C}{\partial \varphi}\right|_{\varphi^*} = 0 \tag{3}$$

This $\varphi^*$ is used to compute the loss Eq. (2) to minimize image and label matching objective. To propagate derivatives from $\varphi^*$ to the feature images $F_f, F_m$, we invoke the Implicit Function Theorem [54]:

**Theorem 1** *For a function $\varrho : \mathbb{R}^n \times \mathbb{R}^{m_1+m_2} \to \mathbb{R}^n$ that is continuously differentiable, if $\varrho(\varphi^*, F_f, F_m) = 0$ and $\left|\frac{\partial \varrho}{\partial \varphi}\right|\big|_{\varphi^*} \neq 0$, then there exist open sets $U, V_f, V_m$ containing $\varphi^*, F_f, F_m$, and a function $\varphi^*(F_f, F_m)$ defined on these open sets such that $\varrho(\varphi^*(F_f, F_m), F_f, F_m) = 0$.*

Given the Implicit Function Theorem, we write $\varrho(\varphi^*(F_f, F_m), F_f, F_m) = 0$ and differentiate with respect to $F_f$ to obtain:

$$\frac{d\varrho}{dF_f} = \frac{\partial \varrho}{\partial \varphi}\frac{\partial \varphi}{\partial F_f} + \frac{\partial \varrho}{\partial F_f} = 0 \implies \frac{\partial \varphi}{\partial F_f} = -\left(\frac{\partial \varrho}{\partial \varphi}\right)^{-1}\frac{\partial \varrho}{\partial F_f} \tag{4}$$

The gradients of $\varphi$ come from Eq. (2) (i.e. $\frac{\partial T}{\partial \varphi}$), and the gradients of $F_f$ w.r.t. Eq. (2) are obtained as $\frac{\partial T}{\partial F_f} = -\frac{\partial T}{\partial \varphi}\left(\frac{\partial \varrho}{\partial \varphi}\right)^{-1}\frac{\partial \varrho}{\partial F_f}$. The gradients of $F_m$ are obtained similarly.

This design ensures that optimal registration in the feature space corresponds to optimal registration *both* in the image and label spaces. Furthermore, the optimization layer ensures that the $\varphi^*$ is a local minima of this high-fidelity feature matching objective, i.e., the features obtained by the network.

**Jacobian-Free Backprop**   In practice, the Jacobian $\frac{\partial \varrho}{\partial \varphi}$ is expensive to compute, given the high dimensionality of $\varphi$ and $\varrho$. Following [31], we substitute the Jacobian to identity, and compute $\frac{\partial \hat{T}}{\partial F_f} \approx -\frac{\partial T}{\partial \varphi}\frac{\partial \varrho}{\partial F_f}$. This leads to much less memory and stable training dynamics compared to other estimates of Jacobian like phantom gradients, damped unrolling, or Neumann series [35, 34].

### 3.3   Multi-scale optimization

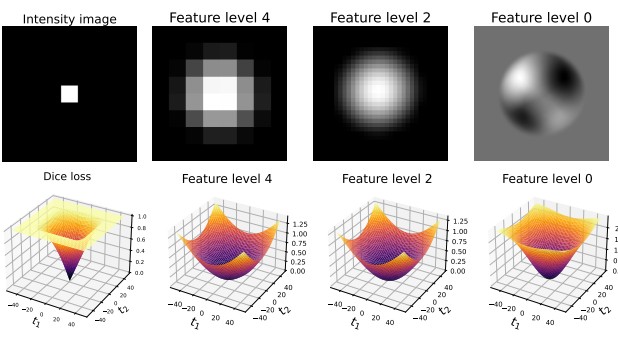

Figure 2: **Dense feature learning leads to flatter loss landscapes**. *Top row* shows the intensity image with the corresponding multi-scale features predicted by the deep network, where the $L^{\text{th}}$ level denotes a feature of size $H/2^k \times W/2^k \times C_k$. *Bottom row* shows the loss landscape as a function of the relative translation between the squares in the fixed and moving image. Note the flat maxima which occurs when there is no overlap between the fixed and moving image, making optimization impossible if there is no overlap of the squares. On the contrary, the loss landscape for learned features is smooth, even at the finest scale, leading to much faster convergence even when there is no overlap between the intensity images.

Optimization based methods typically use a multi-scale approach to improve convergence and avoid local minima with the image matching objective [7, 5, 3, 15]. However, the downsampling of intensity images leads to indiscriminate blurring and loss of details at the coarser scales. We adopt a multi-scale approach by using pyramidal features from the network, which are naturally built into many convolutional architectures. We perform optimization at the coarsest scale, and use the result as initialization for the next finer scale (Algorithm 2). This is similar to optimization methods, but our multi-scale features obtained from different layers in the network correspond to different semantic content, in contrast to classical methods where the multi-scale features are simply downsampled versions of the original images. This allows the multi-scale registration to align different anatomical regions at different scales, which may be hard to align at other finer or coarser scales.

## 4   Experiments

### 4.1   DIO learns dense features from sparse images

A key strength of DIO is the ability to learn interpretable dense features from sparse intensity images for accurate and robust image matching. This is especially relevant for medical image registration, which typically contain a lot of homogenity in the intensity images, making registration difficult. We design a toy task to isolate and demonstrate this behavior. The fixed and moving images are generated by placing a square of size $32\times32$ pixels on an image of $128\times128$ pixels. The squares in

the fixed and moving images overlap with a 50% chance. The task is to find an affine transformation to align the two images. However, classical optimization methods will fail this task 50% of the time, because when the squares do not overlap, there is no gradient of the loss function, illustrated by the flat loss landscape in Fig. 2. However, deep networks discover features that significantly flatten this loss landscape in the feature matching space. To show this, we train a network to output multi-scale feature maps that is used to optimize Eq. (1) to recover an affine transform. We choose a 2D UNet architecture, and the multi-scale feature maps are recovered from different layers of the decoder path of the UNet. Since the features are trained to maximize label matching, the loss landscape is much flatter, and the network is able to recover the affine transform with $> 99\%$ overlap (Appendix A.4). End-to-end learning enables learning of features that are most conducive to registration, unlike existing work [108, 59, 107, 81] that may not contain discriminative registration-aware features about anatomical labels due to lack of task-awareness.

## 4.2 Results on brain MRI registration

**Setup**: We evaluated our method on inter-subject registration on the OASIS dataset [62]. The OASIS dataset contains 414 T1-weighted MRI scans of the brain with label maps containing 35 subcortical structures extracted from automatic segmentation with FreeSurfer and SAMSEG. We use the preprocessed version from the Learn2Reg challenge [42] where all the volumes are skull-stripped, intensity-corrected and center-cropped to $160 \times 192 \times 224$. We use the same training and validation sets as provided in the Learn2Reg challenge to enable fair comparison with other methods.

**Architectures**: We consider four architectures for the task, representing different inductive biases in the network. We use a 3D UNet architecture (denoted as *UNet* in experiments), and a large-kernel UNet (denoted as *LKU*) [48]. To extract multi-scale features from the networks, we attach single convolutional layers to the feature of the desired scales from the decoder path. For each of these architectures, we also consider "Encoder-Only" versions by discarding the decoder path, and creating independent encoders for each scale Fig. 9, denoted as *UNet-E* and *LKU-E*. We choose Encoder-Only versions to ablate the performance using shared features from the decoder path versus independent feature extraction at each scale.

**Results**: We compare our method with existing methods on the Learn2Reg OASIS challenge (Table 1). We compare with state-of-the-art classical methods [5, 46, 64, 100], and deep networks [58, 87, 67, 14, 22, 48]. DIO is highly competitive with existing methods,

Table 1: **Performance on OASIS validation set.** DIO is highly competitive with state-of-the-art DLIR methods in the in-distribution setting. Our feature learning incorporates label-aware features, which is evident from the superior performance compared to four SOTA optimization-based classical methods.

| Validation | | |
|---|---|---|
| **Method** | **Dice** | **HD95** |
| ANTs [5] | $0.786 \pm 0.033$ | $2.209 \pm 0.534$ |
| NiftyReg [64] | $0.775 \pm 0.029$ | $2.382 \pm 0.723$ |
| LogDemons [100] | $0.804 \pm 0.022$ | $2.068 \pm 0.448$ |
| FireANTs [46] | $0.791 \pm 0.028$ | $2.793 \pm 0.602$ |
| Progressive C2F [58] | $0.827 \pm 0.013$ | $1.722 \pm 0.318$ |
| Little learning[87] | $0.846 \pm 0.016$ | $1.500 \pm 0.304$ |
| CLapIRN [67] | $0.861 \pm 0.015$ | $1.514 \pm 0.337$ |
| Voxelmorph-huge [14] | $0.847 \pm 0.014$ | $1.546 \pm 0.306$ |
| TransMorph [22] | $0.858 \pm 0.014$ | $1.494 \pm 0.288$ |
| TransMorph-Large [22] | $0.862 \pm 0.014$ | $1.431 \pm 0.282$ |
| Ours (UNet-E) | $0.845 \pm 0.018$ | $1.790 \pm 0.433$ |
| Ours (LKU-E) | $0.849 \pm 0.018$ | $1.733 \pm 0.401$ |
| Ours (UNet) | $0.853 \pm 0.018$ | $1.675 \pm 0.379$ |
| Ours (LKU) | $0.862 \pm 0.017$ | $1.584 \pm 0.351$ |

especially with TransMorph which uses up to two orders of magnitude more trainable parameters than DIO to achieve a similar performance. We note that the Large Kernel UNet architecture performs better than the standard UNet architecture, which is consistent with the findings in [48], even for dense feature extraction. This is due to the larger receptive field of LKUNet, which is able to capture more context in the image. Moreover, the Encoder-Only versions of the network perform slightly worse than the full networks, showing that sharing features across scales is beneficial for the task.

## 4.3 Optimization-in-the-loop introduces robustness to domain shift

A key requirement of registration algorithms is to generalize over a spectrum of acquisition and preprocessing protocols, since medical images are rarely acquired with the same configuration. Existing DLIR methods are extremely sensitive to domain shift, and catastrophically fail on other brain datasets. On the contrary, DIO inherits the domain agnosticism of the optimization solver, and is robust under feature distortions introduced by domain shift.

We evaluate the robustness of the trained models on three brain datasets: LPBA40, IBSR18, and CUMC12 datasets [85, 1, 53]. Contrary to the OASIS dataset, these datasets were obtained on

different scanners, aligned to different atlases (MNI305, Talairach) with varying algorithms used for skull-stripping, bias correction (BrainSuite, autoseg), and different manual labelling protocols of different anatomical regions (as opposed to automatically generated Freesurfer labels in OASIS). Unlike the OASIS dataset, these datasets have different volume sizes, and IBSR18 and CUMC12 datasets are not 1mm isotropic. More details about the datasets are provided in Appendix A.6.

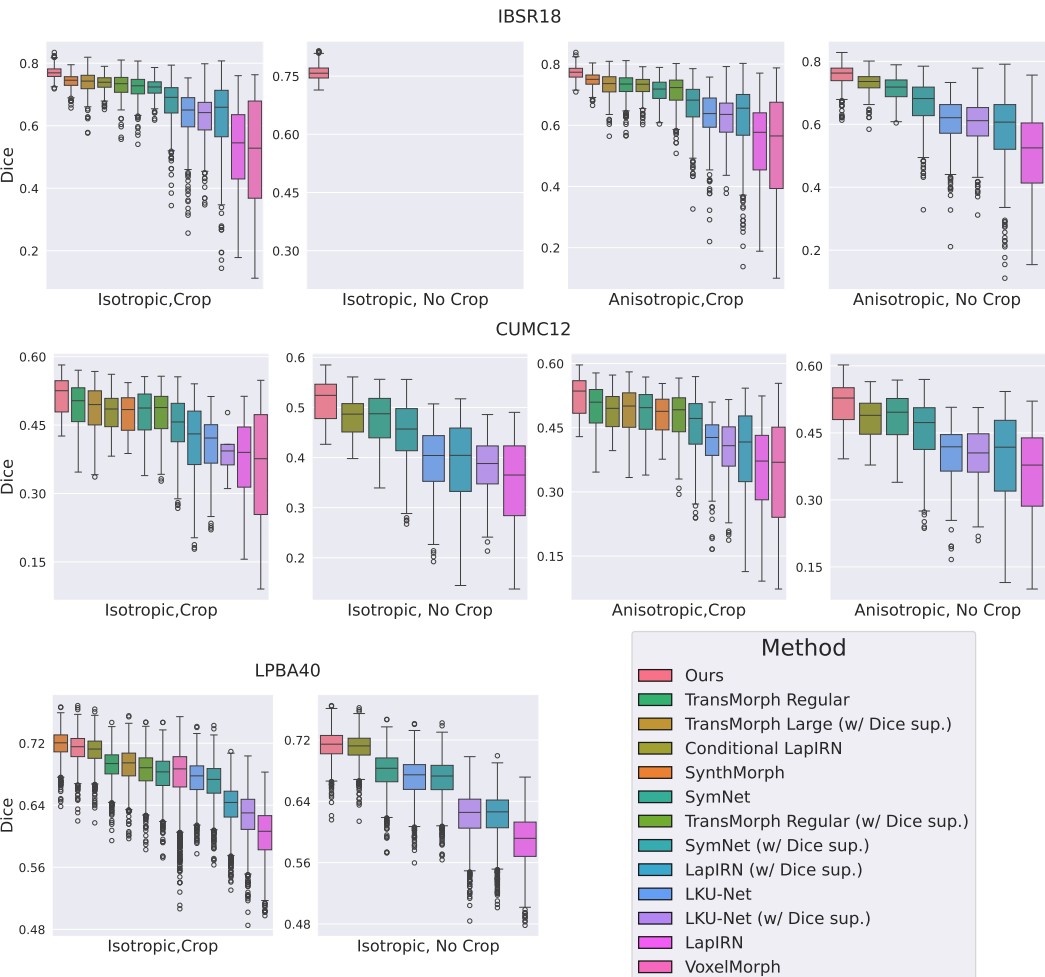

Figure 3: **Boxplots of Dice scores for three out-of-distribution datasets.** DIO performs significantly better across three datasets without additional finetuning. Contrary to other baselines that output warp fields considering 1mm isotropic data, leading to a performance drop with anisotropic volumes, DIO performs better with anisotropic data due to the optimization's resolution-agnostic nature.

**Results**. We evaluate across a variety of configurations – (i) preserving the anisotropy of the volumes or resampling to 1mm isotropic (denoted as *anisotropic* or *isotropic*), and (ii) center-cropping the volumes to match the size of the OASIS dataset (denoted as *Crop* and *No Crop*). The results for all three datasets are shown in Fig. 3 sorted by mean Dice score; quantitative comparison is also shown in Appendix Table 4. Note that TransMorph, VoxelMorph, and SynthMorph do not work for sizes that are different than the OASIS dataset, therefore they only work in the *Crop* setting. The IBSR18 dataset also has volumes with different spatial sampling, and resampling to 1mm isotropic leads to different voxel sizes. These volumes cannot be concatenated along the channel dimension, consequently every DLIR method cannot run under this configuration (Fig. 3(a)). Since our method takes as input only a single volume, and the convolutional architecture preserves the volume size, the fixed and moving images can have different voxel sizes, i.e. feature extraction is not contingent on the voxel sizes of the moving and fixed images being equal. The optimization solver can also handle different voxel sizes for the fixed and moving volumes – which is useful in applications like multimodal registration (in-vivo to ex-vivo, histology to 3D, MRI to microscopy). This unprecedented flexibility brings forth

a new operational paradigm in deep learning for registration that was unavailable before, widening the scope of applications for registration with deep features.

We compare our method with a variety of DLIR baselines, trained with and without label supervision (the former denoted as '*w/ Dice sup.*' in Fig. 3). Our method performs substantially better than all the baselines with a significantly narrower interquartile range on the IBSR18 and CUMC12 datasets. The differences are significant – on IBSR18 and CUMC12, our median performance is higher than the third quartile of almost all baselines. The sturdy performance against domain shift provides a strong motivation for using optimization-in-the-loop for learnable registration.

## 4.4 Robust feature learning enables zero-shot performance by switching optimizers at test-time

Another major advantage of our framework is that we can switch the optimizer *at test time* without any retraining. This is useful when the registration constraints change over time (i.e. initially diffeomorphic transforms were required but now non-diffeomorphic transforms are acceptable), or when the registration is used in a pipeline where different parameterizations (freeform, diffeomorphic, geodesic, B-spline) may be compared. Since our framework decouples the feature learning from the optimization, we can switch the optimizer arbitrarily at test time, at no additional cost. A crucial requirement is that learned features should not be too sensitive to the training optimizer.

| Optimizer | SGD | | | FireANTs (diffeomorphic) | | |
|---|---|---|---|---|---|---|
| Architecture | DSC | HD95 | $\%(\|\mathbf{J}\| < 0)$ | DSC | HD95 | $\%(\|\mathbf{J}\| < 0)$ |
| UNet Encoder | $0.845 \pm 0.018$ | $1.790 \pm 0.433$ | $0.7866 \pm 0.1371$ | $0.834 \pm 0.018$ | $1.847 \pm 0.410$ | $0.0000 \pm 0.0000$ |
| LKU Encoder | $0.849 \pm 0.018$ | $1.733 \pm 0.401$ | $0.8079 \pm 0.1308$ | $0.838 \pm 0.018$ | $1.806 \pm 0.373$ | $0.0000 \pm 0.0000$ |
| UNet | $0.853 \pm 0.018$ | $1.675 \pm 0.379$ | $1.0718 \pm 0.1662$ | $0.842 \pm 0.018$ | $1.748 \pm 0.397$ | $0.0000 \pm 0.0000$ |
| LKU | $0.862 \pm 0.017$ | $1.584 \pm 0.351$ | $0.8646 \pm 0.1429$ | $0.849 \pm 0.017$ | $1.740 \pm 0.345$ | $0.0000 \pm 0.0000$ |

Table 2: **Zero shot performance by switching optimizers at test-time**. Our method is trained on the OASIS dataset with the SGD optimizer to obtain the warp field. At inference time, we use an SGD optimizer for no constraint on the warp field, and the FireANTs optimizer to ensure diffeomorphic warps. Across all architectures, the Dice Score remains robust, with only a slight dip attributed to the constraints introduced by diffeomorphic mappings. The SGD optimization introduces ∼1% singularities, while FireANTs shows no singularities.

To demonstrate this functionality, we use the validation set of the OASIS dataset and the four networks trained in Section 4.2. The networks were initially trained on the SGD optimizer without any additional constraints on the warp field. At test time, we switch the optimizer to the FireANTs optimizer [46], that uses a Riemannian Adam optimizer for multi-scale diffeomorphisms. Results in Table 2 compare the Dice score, 95th percentile of the Haussdorf distance (denoted as *HD95*) and percentage of volume with negative Jacobians (denoted as $\%(\|J\| < 0)$) for the two optimizers. The SGD optimizer introduces anywhere from $0.79\%$ to $1.1\%$ of singularities in the registration, while the FireANTs optimizer does not introduce any singularities. A slight drop in performance can be attributed to the additional constraints imposed by diffeomorphic transforms. However, the high-fidelity features lead to a much better label overlap than FireANTs run with image features (Table 1). Our framework introduces an unprecedented amount of flexibility at test time that is an indispensible feature in deep learning for registration, and can be useful in a variety of applications where the registration requirements change over time, without expensive retraining.

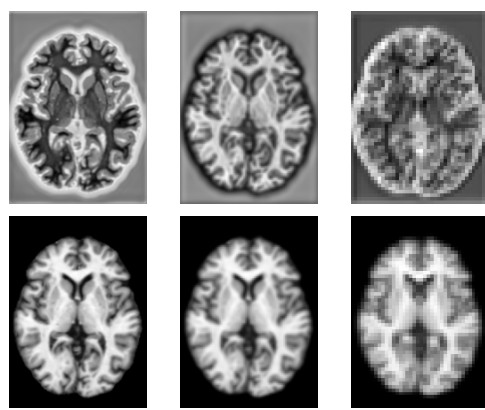

Figure 4: Examples of multi-scale features learned by the feature extractor. Scale-space features (*bottom row*) obtained by downsampling the image downsample all image features indiscriminately. Our features (*top row*) preserve necessary anatomical information at all scales, and introduce inhomogenity in the feature space for better optimization (watershed effect and enhanced contrast near gyri and a halo around the outer surface to delineaate background from gray matter).

## 4.5 Interpretability of features

Decoupling of feature learning and optimization allows us to examine the feature images obtained at each scale to understand what feature help in the registration task. Classical methods use scale-space images (smoothened and downsampled versions of the original image) to avoid local minima, but lose discriminative image features at lower resolutions. Moreover, intensity images may not provide sufficient details to perform label-aware registration. Since our method learns dense features to minimize label matching losses, we can observe which features are necessary to enable label-aware registration. Fig. 4 highlights differences between scale-space images and features learned by our network. At all scales, the features introduces heterogeneity using a watershed effect and enhanced contrast to improve label matching performance.

## 4.6 Inference time

DLIR methods have been very popular due to their fast inference time by performing amortized optimization [14]. Classical methods generally focus on robustness and reproducibility, and do have GPU implementations for fast inference. However, modern optimization toolkits [60, 46] utilize massively parallel GPU computing to register images in seconds, and scale very well to ultrahigh resolution imaging. A concern with optimization-in-the-loop methods is the inference time. Table Table 3 shows the inference time for our method for all four architectures. These inference times are fast for a lot of applications, and the plug-and-play nature of our framework makes DIO amenable to rapid experimentation and hyperparameter tuning.

# 5 Conclusion and Limitations

| Architecture | Neural net | Optimization |
|---|---|---|
| UNet | 0.444 | 1.693 |
| UNet-E | 0.433 | 1.555 |
| LKU | 0.795 | 1.463 |
| LKU-E | 2.281 | 1.457 |

Table 3: **Inference time for various architectures**. A multi-scale optimization takes only $\sim 1.5$ seconds to run all iterations (no early stopping) making it suitable for most applications. This is compared to the time for neural network's feature extraction which is architecture dependent.

**Conclusion**   DLIR methods provide several benefits such as amortized optimization, integration of weak supervision, and the ability to learn from large (labeled) datasets. However, coupling of the feature learning and optimization steps in DLIR methods limits the flexibility and robustness of the deep networks. In this paper, we we introduce a novel paradigm that incorporates optimization-as-a-layer for learning-based frameworks. This paradigm retains all the flexibility and robustness of classical multi-scale methods while leveraging large scale weak supervision such as anatomical landmarks into *high-fidelity, registration-aware feature learning*. Our paradigm allows "promptable" registration out-of-the-box as part of the plug-and-play optimization, where additional supervision such as labelmaps or landmarks can be added to the optimization loss at test time. Our fast implementation allows for implementation of optimization-as-a-layer in deep learning, which was previously thought to be infeasible, due to existing optimization frameworks being prohibitively slow. Densification of features from our method also leads to better optimization landscapes, and our method is robust to unseen anisotropy and domain shift. To our knowledge, our method is the first to switch between transformation representations (free-form to diffeomorphic) at *test time* without any retraining. This comes with fast inference runtimes, and interpretability of the features used for optimization. Potential future work can explore multimodal registration, online hyperparameter tuning and few-shot learning.

**Limitations**   The first limitation is unlike existing DLIR methods that concatenate the fixed and moving images to feed into the network, DIO processes the images independently. The features extracted from an image are therefore trained to marginalize the label matching performance over all possible moving images, and cannot adapt to the moving image. This leads to slightly asymptotically lower in-domain performance than methods like [48]. The second limitation is the implicit bias of the optimization algorithm. Implicit bias in SGD restricts the space of solutions for optimization problems that are overparameterized, such as deep networks [113, 90, 47, 74, 109]. In deformable registration, the implicit bias of SGD restricts the direction of the gradient of the particle at $\varphi(x)$, which is *always parallel* to $\nabla F_m(\varphi(x))$, independent of the fixed image and dissimilarity function. This limits the degrees of freedom of the optimization by $N$-fold for $N$-D images. This is unlike DLIR methods where the warp is not constrained to move along $\nabla F_m(\varphi(x))$. This behavior is explored in more detail in Appendix A.1. Future work aims to mitigate this implicit bias for better performance.

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

## A   Appendix

### A.1   Implicit bias of optimization for registration

Model based systems, such as deep networks are not immune to inductive biases due to architecture, loss functions, and optimization algorithms used to train them. Functional forms of the deep network induce constraints on the solution space, but optimization algorithms are not excluded from such biases either. The implicit bias for Gradient Descent is a well-studied phenomena for overparameterized linear and shallow networks. Gradient Descent for linear systems leads to an optimum that is in the span of the input data starting from the initialization [113, 90, 47, 74, 109]. This bias is also dependent on the chosen representation, since that defines the functional relationship of the gradients with the parameters and inputs. This limits the reachable set of solutions by the optimization algorithm when multiple local minima exist.

In the case of image registration, the optimization limits the space of solutions (warps) that can be obtained by the SGD algorithm. To show this, we consider the transformation $\varphi$ as a set of particles in a Langrangian frame that are displaced by the optimization algorithm to align the moving image to the fixed image. Consider a regular grid of particles, whose locations specify the warp field. Let the location of $i$-th particle at iteration $t$ be $\varphi^{(t)}(\mathbf{x}_i)$. For a fixed feature image $F_f$, moving image $F_m$ and current iterate $\varphi^{(t)}$, the gradient of the registration loss with respect to particle $i$ at iteration $t$ is given by

$$\frac{\partial C(F_f, F_m \circ \varphi^{(t)})}{\partial \varphi^{(t)}(\mathbf{x}_i)} = C_i'(F_f, F_m \circ \varphi^{(t)}) \nabla F_m(\varphi^{(t)}(\mathbf{x}_i)) \tag{5}$$

where

$$C_i'(F_f, F_m \circ \varphi^{(t)}) = \frac{\partial C(F_f, F_m \circ \varphi^{(t)})}{\partial M(\varphi^{(t)}(\mathbf{x}_i))}$$

is the (scalar) derivative of scalar loss $C$ with respect to the intensity of $i$-th particle computed at the current iterate, and $\nabla F_m(\varphi^{(t)}(\mathbf{x}_i))$ is the spatial gradient of the moving image at the location of the particle. Note that the **direction** of the gradient of particle $i$ is *independent* of the fixed image, loss function, and location of other particles – it only depends on the spatial gradient of the moving image at the location of the particle. This restricts the movement of a particle located at any given location along a 1D line whose direction is the spatial gradient of the moving image at that location. Since $F_f$ and $F_m$ are computed independently of each other (and therefore no information of $F_f$ and $F_m$ is contained in each other), the space of solutions of $\varphi$ is restricted by this implicit bias. This is restrictive because the similarity function and fixed image do not influence the direction of the gradient, and the optimization algorithm is biased towards solutions that are in the direction of the gradient of the moving image.

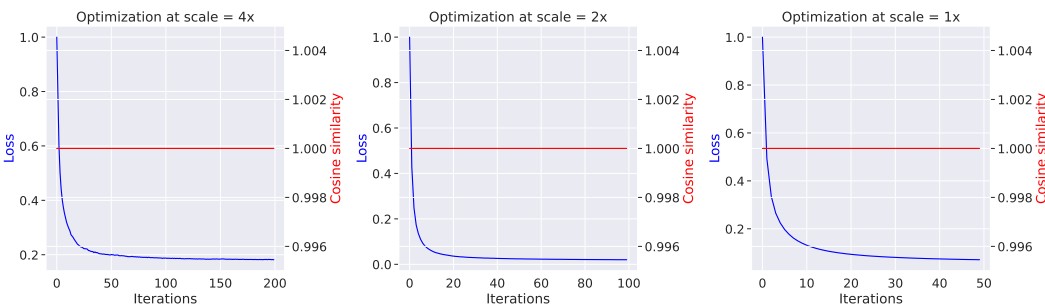

Figure 5: **Implicit bias in SGD for image registration.** The plot shows the loss curves for a multi-scale optimization of two feature images. Each plot also shows the absolute cosine similarity of per-pixel gradients obtained by $C$ and $C_{\text{surrogate}}$ at each iteration. Note that over the course of optimization, the cosine similarity is always 1 – demonstrating the implicit bias of the optimization for registration.

We show this bias empirically – we perform multi-scale optimization algorithm using feature maps obtained from the network. We keep track of two gradients, one obtained by the loss function, and another obtained by the gradient of a surrogate loss $C_{\text{surrogate}}(F_m, \varphi^{(t)}) = \sum_i F_m(\varphi^{(t)}(\mathbf{x}_i))$. Note that $C_{\text{surrogate}}$ does not depend on the fixed image or the loss function. The gradient of $C_{\text{surrogate}}$ with

respect to the $i$-th particle is given by $\nabla F_m(\varphi^{(t)}(\mathbf{x}_i))$. At each iteration, we compute the magnitude of cosine similarly between the gradients of $C$ and $C_{\text{surrogate}}$. Fig. 5 shows that the loss converges, and the per-pixel gradients can be predicted by $C_{\text{surrogate}}$ alone, as depicted by the magnitude and standard deviation of cosine similarity between $C$ and $C_{\text{surrogate}}$. This limits the movement of each particle along a 1D line in an $N$-D space, and limits the degrees of freedom of the optimization by $N$-fold for $N$-D images. Future work will aim at alleviating this implicit bias to allow for more flexible solutions.

## A.2 Algorithm details

DIO is a learnable framework that leverages *implicit differentiation* of an arbitrary black-box optimization solver to learn features such that registration in this feature space corresponds to good registration of the images and additional label maps. This additional indirection leads to learnable features that are registration-aware, interpretable, and the framework inherits the optimization solver's versatility to variability in the data like difference in contrast, anisotropy, and difference in sizes of the fixed and moving images. We contrast our approach with a typical classical optimization-based registration algorithm in Fig. 6. A classical multi-scale optimization routine *indiscriminately* downsamples the intensity images, and does not retain discriminative information that is useful for registration. Since our method is trained to maximize label alignment from all scales, multi-scale features obtained from our method are more discriminative and registration-aware. We also compare DIO with a typical DLIR method in Fig. 7. Note that the fixed end-to-end architecture and functional form of a deep network subsumes the representation choice into the architecture as well, limiting its ability to switch to arbitrary transformation representations at inference time without additional retraining. Our framework therefore combines the benefits of both classical (robustness to out-of-distribution datasets, and zero-shot transfer to other optimization routines) and learning-based methods (high-fidelity, label-aware, and registration-aware).

## A.3 Implementation Details

For all experiments, we use downsampling scales of $1, 2, 4$ for the multi-scale optimization. All our methods are implemented in PyTorch, and use the Adam optimizer for learning the parameters of the feature network. Note that in Eq. (3), $\varrho$ is the partial derivative of the loss function $C$ with respect to the transformation $\varphi$, which contains a $\nabla(F_m \circ \varphi)$ term, which is the backward transform of the `grid_sample` operator in PyTorch. Since this operation is not implemented using PyTorch primitives, a backward pass for the gradient operation does not exist in PyTorch. We use the `gridsample_grad2` library [86] to compute the gradients of the backward pass of the `grid_sample` operator, used in Eq. (3). All experiments are performed on a single NVIDIA A6000 GPU.

## A.4 Toy example

Fig. 8 shows the loss curves for the toy dataset described in Section 4.1. An image-based optimization algorithm would correspond to the green curve being a flat line at $1$ due to the flat landscape of the intensity-based loss function.

## A.5 Quantitative Results

Table 4 shows the quantitative results of our method for out-of-distribution performance on the IBSR18, CUMC12, and LPBA40 datasets. In 9 out of 10 cases, DIO demonstrates the best accuracy with fairly lower standard deviations, highlighting the robustness of the model. DIO therefore serves as a strong candidate for out-of-distribution performance, and can be used in a variety of settings where the training and test distributions differ.

## A.6 Datasets

We consider four brain MRI datasets in this paper: OASIS dataset for in-distribution performance, and LPBA40, IBSR18, and CUMC12 datasets for out-of-distribution performance [85, 1, 53, 62]. More details about the datasets are provided below.

- **OASIS**. The Open Access Series of Imaging Studies (OASIS) dataset contains 414 T1-weighted brain images in Young, Middle Aged, Nondemented, and Demented Older adults. The images are skull-stripped and bias-corrected, followed by a resampling and afine alignment to the FreeSurfer's Talairach atlas. Label segmentations of 35 subcortical structures were obtained using automatic segmentation using Freesurfer software.

---

**Algorithm 1** Classical registration pipeline

---

1: **Input:** Fixed image $I_f$, Moving image $I_m$
2: Scales $[s_1, s_2, \ldots, s_n]$, Iterations $[T_1, T_2, \ldots T_n]$, $n$ levels.
3: Initialize $\varphi = \mathbf{Id}_{s_1}$.            ▷ Initialize warp to identity at first scale
4: Initialize $l = 1$.            ▷ Initialize current scale
5: **while** $l \leq n$ **do**
6:      Initialize $i = 0$
7:      Initialize $I_f^l, I_m^l = \text{downsample}(I_f, s_l), \text{downsample}(I_m, s_l)$
8:      **while** $i < T_l$ **do**
9:          $L_i = C(I_f^l, I_m^l \circ \varphi^i)$
10:          Compute $\nabla_\varphi L$
11:          Update $\varphi^{(i+1)} = \text{Optimize}(\varphi^i, \nabla_\varphi L_i)$      ▷ Optimization algorithm
12:          $i = i + 1$
13:      **end while**
14:      **if** $l < n$ **then**
15:          $\varphi = \text{Upsample}(\varphi, s_{(l+1)})$      ▷ Upsample warp to next level
16:      **end if**
17:      $l = l + 1$
18: **end while**

---

---

**Algorithm 2** Differentiable Implicit Optimization for Registration (Our algorithm)

---

1: **Input:** Fixed features $\mathcal{F}_f = [F_f^1, F_f^2 \ldots F_f^n]$, Moving features $\mathcal{F}_f = [F_f^1, F_f^2 \ldots F_f^n]$
2: Scales $[s_1, s_2, \ldots, s_n]$, Iterations $[T_1, T_2, \ldots T_n]$, $n$ levels.
3: Initialize $\varphi = \mathbf{Id}_{s_1}$.      ▷ Initialize warp to identity at first scale
4: Initialize $l = 1$.      ▷ Initialize current scale
5: Outputs $= []$.      ▷ Save intermediate outputs for backpropagation
6: **while** $l \leq n$ **do**
7:      Initialize $i = 0$
8:      Initialize $I_f^l, I_m^l = F_f^l, F_m^l$
9:      **while** $i < T_l$ **do**
10:          $L_i = C(I_f^l, I_m^l \circ \varphi^i)$
11:          Compute $\nabla_\varphi L$
12:          Update $\varphi^{(i+1)} = \text{Optimize}(\varphi^i, \nabla_\varphi L_i)$      ▷ Optimization algorithm
13:          $i = i + 1$
14:      **end while**
15:      Outputs.append $\left( \varphi^{(T_l)} \right)$      ▷ Save final warp at this level for backpropagation
16:      **if** $l < n$ **then**
17:          $\varphi = \text{Upsample}(\varphi, s_{(l+1)})$      ▷ Upsample warp for next level
18:      **end if**
19:      $l = l + 1$
20: **end while**

---

Figure 6: **Comparison of a typical classical registration algorithm and DIO:** Algorithm 1 shows a typical classical registration algorithm that uses a multi-scale optimization routine to register the fixed and moving images. At each level $l$, the fixed and moving images are downsampled by a factor of $s_l$, therefore trading off between discriminative information and vulnerability to local minima. Algorithm 2 shows our algorithm (red text highlights differences compared to Algorithm 1) that uses a separate scale-space feature at each level. Unlike classical methods, the scale-space feature can capture different discriminative features at each level to maximize label alignment and the multi-scale nature helps avoid local minima.

- **LPBA40**. 40 brain images and their labels are used to construct the LONI Probabilistic Brain Atlas (LPBA40) dataset at the Laboratory of Neuroimaging (LONI) at UCLA [85]. All volumes are preprocessed according to LONI protocols to produce skull-stripped volumes. These volumes are aligned to the MNI305 atlas – this is relevant since existing DLIR methods may be biased towards

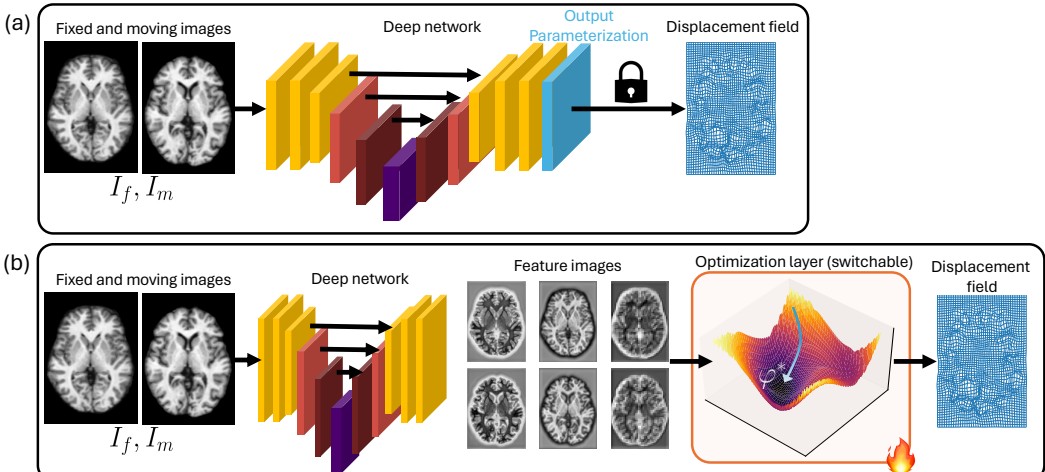

Figure 7: **Comparison of typical DLIR method and our method. (a)** shows the pipeline of a typical deep network. The neural network architecture takes the channelwise concatenation of the fixed and moving images as input, and outputs a warp field, which has a *fixed* transformation representation (SVF, free-form, B-splines, affine, etc. denoted as the blue locked layer). This representation is fixed throughout training and cannot be switched at test-time, without additional finetuning of the network. **(b)** shows our framework wherein the fixed and moving images are input *separately* into a feature extraction network that outputs multi-scale features. These features are then passed onto an iterative black-box solver than can be *implicitly differentiated* to backpropagate the gradients from the optimized warp field back to the feature network. This allows for a more flexible transformation representation, and the optimization solver can be switched at test-time with zero finetuning.

814     images that are aligned to the Talairach and Tournoux (1988) atlas which is used to align the images
815     in the OASIS dataset. This is followed by a custom manual labelling protocol of 56 structures from
816     each of the volumes. Bias correction is perfrmed using the BrainSuite's Bias Field Corrector.

817 • **IBSR18**. the Internet Brain Segmentation Repository contains 18 different brain images acquired
818     at different laboratories as IBSRv2.0. The dataset consists of T1-weighted brains aligned to the
819     Talairach and Tournoux (1988) atlas, and manually segmented into 84 labelled regions. Bias
820     correction of the images are performed using the 'autoseg' bias field correction algorithm.

821 • **CUMC12**. The Columbia University Medical Center dataset contains 12 T1-weighted brain images
822     with manual segmentation of 128 regions. The images were scanned on a 1.5T GE scanner, and the
823     images were resliced coronally to a slice thickness of 3mm, rotated into cardinal orientation, and
824     segmented by a technician trained according to the Cardviews labelling scheme.

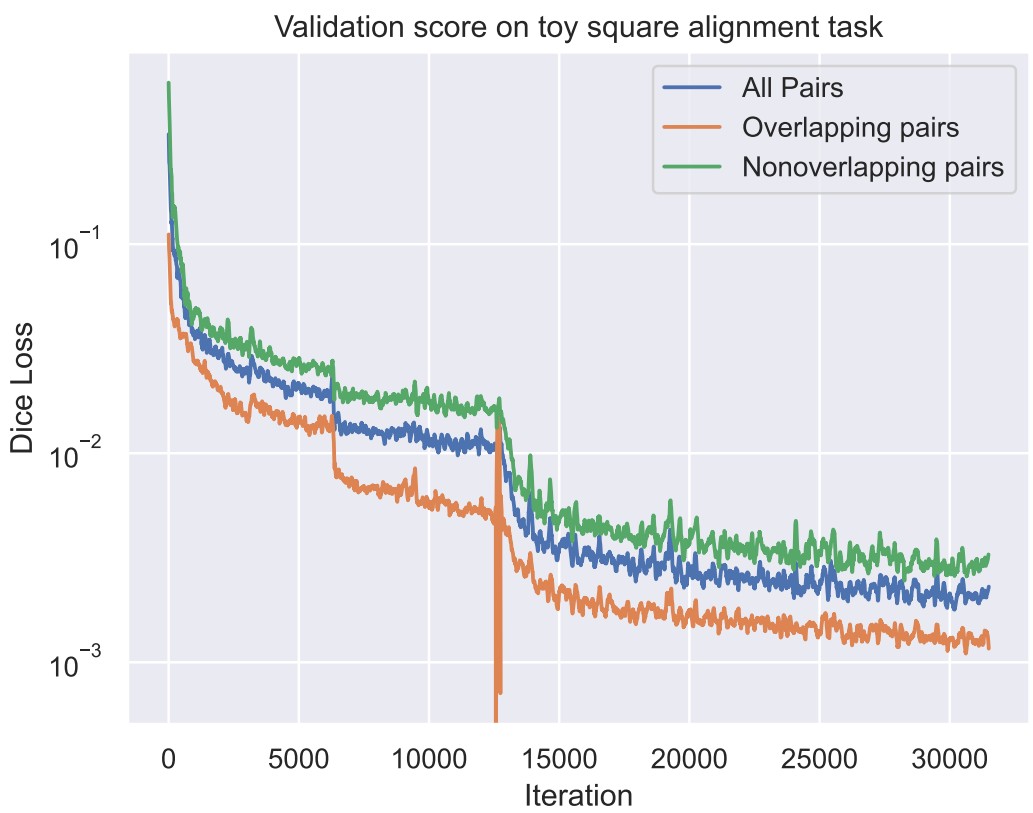

Figure 8: **Loss curves for toy dataset**. Plot shows three curves - the Dice score for (a) all validation image pairs, (b) image pairs that have non-zero overlap in the image space (therefore a gradient-based affine solver will recover a transform from intensity images), and (c) image pairs that have zero overlap in the image space (therefore any gradient-based solver using intensity images will fail). Our feature network recovers dense multi-scale features (see Fig. 2) which allows all subsets to be registered with >0.99 Dice score.

| Method | Dice supervision | Isotropic | | Anisotropic | |
|---|---|---|---|---|---|
| | | **Crop** | **No Crop** | **Crop** | **No Crop** |
| Conditional LapIRN | ✗ | 0.7367 ± 0.0237 | ✗ | 0.7269 ± 0.0328 | 0.7317 ± 0.0303 |
| LapIRN | ✗ | 0.5257 ± 0.1316 | ✗ | 0.5435 ± 0.1266 | 0.5001 ± 0.1271 |
| LapIRN | ✓ | 0.6259 ± 0.1238 | ✗ | 0.6209 ± 0.1163 | 0.5759 ± 0.1207 |
| LKU-Net | ✗ | 0.6309 ± 0.0839 | ✗ | 0.6276 ± 0.0838 | 0.6072 ± 0.0787 |
| LKU-Net | ✓ | 0.6267 ± 0.0776 | ✗ | 0.6231 ± 0.0730 | 0.5992 ± 0.0757 |
| SymNet | ✗ | 0.7213 ± 0.0273 | ✗ | 0.7116 ± 0.0398 | 0.7117 ± 0.0398 |
| SymNet | ✓ | 0.6731 ± 0.0688 | ✗ | 0.6672 ± 0.0731 | 0.6674 ± 0.0728 |
| TransMorph Large | ✓ | 0.7383 ± 0.0353 | ✗ | 0.7312 ± 0.0405 | ✗ |
| TransMorph Regular | ✗ | 0.7221 ± 0.0400 | ✗ | 0.7289 ± 0.0417 | ✗ |
| TransMorph Regular | ✓ | 0.7293 ± 0.0370 | ✗ | 0.7113 ± 0.0520 | ✗ |
| VoxelMorph | ✗ | 0.5118 ± 0.1774 | ✗ | 0.5233 ± 0.1693 | ✗ |
| SynthMorph | ✓ | 0.7423 ± 0.0225 | ✗ | 0.7476 ± 0.0238 | ✗ |
| Ours (LKU) | ✓ | 0.7698 ± 0.0193 | 0.7587 ± 0.0208 | 0.7728 ± 0.0219 | 0.7572 ± 0.0369 |
| Conditional LapIRN | ✗ | 0.4793 ± 0.0373 | 0.4804 ± 0.0368 | 0.4880 ± 0.0416 | 0.4827 ± 0.0408 |
| LapIRN | ✗ | 0.3719 ± 0.0897 | 0.3491 ± 0.0895 | 0.3524 ± 0.1001 | 0.3556 ± 0.0989 |
| LapIRN | ✓ | 0.4121 ± 0.0907 | 0.3838 ± 0.0929 | 0.3911 ± 0.1060 | 0.3896 ± 0.1063 |
| LKU-Net | ✗ | 0.4054 ± 0.0641 | 0.3922 ± 0.0679 | 0.4086 ± 0.0732 | 0.3999 ± 0.0697 |
| LKU-Net | ✓ | 0.3904 ± 0.0547 | 0.3827 ± 0.0574 | 0.3967 ± 0.0745 | 0.3960 ± 0.0678 |
| SymNet | ✗ | 0.4761 ± 0.0524 | 0.4761 ± 0.0524 | 0.4822 ± 0.0565 | 0.4820 ± 0.0565 |
| SymNet | ✓ | 0.4457 ± 0.0675 | 0.4457 ± 0.0675 | 0.4518 ± 0.0787 | 0.4521 ± 0.0786 |
| TransMorph Large | ✓ | 0.4827 ± 0.0531 | ✗ | 0.4858 ± 0.0587 | ✗ |
| TransMorph Regular | ✗ | 0.4929 ± 0.0502 | ✗ | 0.4967 ± 0.0540 | ✗ |
| TransMorph Regular | ✓ | 0.4737 ± 0.0549 | ✗ | 0.4741 ± 0.0628 | ✗ |
| VoxelMorph | ✗ | 0.3519 ± 0.1271 | ✗ | 0.3469 ± 0.1308 | ✗ |
| SynthMorph | ✓ | 0.4761 ± 0.0397 | ✗ | 0.4797 ± 0.0426 | ✗ |
| Ours (LKU) | ✓ | 0.5137 ± 0.0410 | 0.5126 ± 0.0412 | 0.5237 ± 0.0433 | 0.5162 ± 0.0448 |
| Conditional LapIRN | ✗ | 0.7113 ± 0.0178 | 0.7109 ± 0.0178 | - | - |
| LapIRN | ✗ | 0.6026 ± 0.0317 | 0.5878 ± 0.0325 | - | - |
| LapIRN | ✓ | 0.6395 ± 0.0269 | 0.6211 ± 0.0294 | - | - |
| LKU-Net | ✗ | 0.6746 ± 0.0230 | 0.6708 ± 0.0249 | - | - |
| LKU-Net | ✓ | 0.6266 ± 0.0299 | 0.6220 ± 0.0296 | - | - |
| SymNet | ✗ | 0.6797 ± 0.0239 | 0.6797 ± 0.0238 | - | - |
| SymNet | ✓ | 0.6700 ± 0.0248 | 0.6698 ± 0.0248 | - | - |
| TransMorph Large | ✓ | 0.6918 ± 0.0219 | ✗ | - | - |
| TransMorph Regular | ✗ | 0.6919 ± 0.0191 | ✗ | - | - |
| TransMorph Regular | ✓ | 0.6855 ± 0.0225 | ✗ | - | - |
| VoxelMorph | ✗ | 0.6776 ± 0.0365 | ✗ | - | - |
| SynthMorph | ✓ | 0.7189 ± 0.0172 | ✗ | - | - |
| Ours (LKU) | ✓ | 0.7139 ± 0.0181 | 0.7131 ± 0.0181 | - | - |

Table 4: **Quantitative evaluation on out-of-distribution performance on IBSR18, CUMC12, and LPBA40 datasets.** We compare DIO with other state-of-the-art DLIR methods. The '**Dice supervision**' column shows if the method is trained with label matching on the OASIS dataset. We evaluate the performance of the methods with and without isotropic and anisotropic data resampling. The results are reported as mean ± standard deviation. ▮ = First, ▮ = Second, ▮ = Third best result.

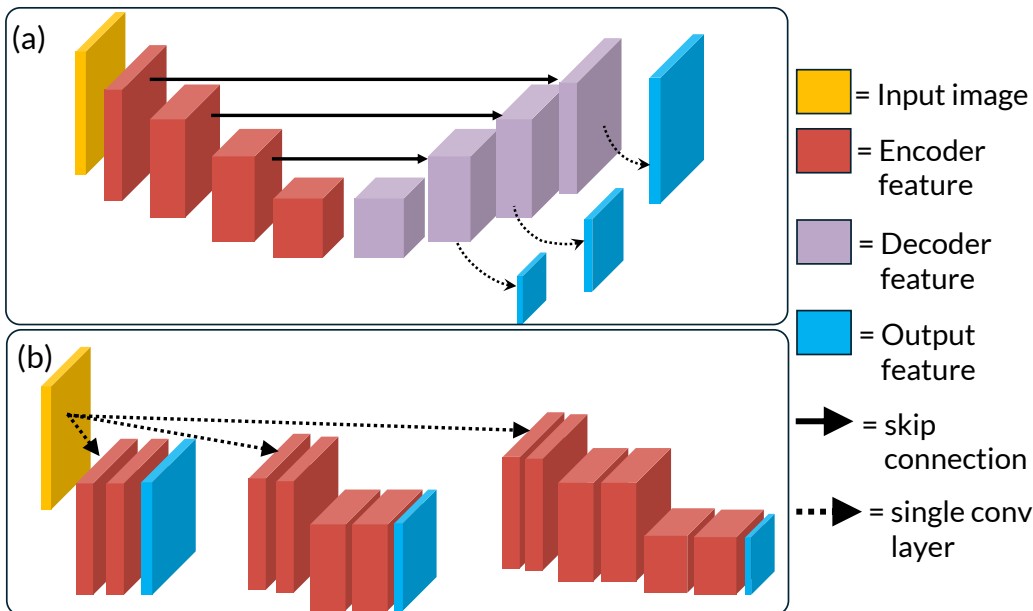

Figure 9: **Architecture details**. **(a)** illustrates the UNet and Large Kernel U-Net (LKUNet) architecture designs, which consists of encoder blocks (red) and decoder blocks (purple) linked using skip connections. Multi-scale features are extracted from the intermediate decoder layers using a single convolutional layer. This design leads to shared features across multiple scales. UNet and LKUNet differ in the kernel parameters within each encoder and decoder blocks. **(b)** illustrates the 'Encoder-Only' versions of the same networks. The decoder path is entirely discarded, and each feature image is extracted using a separate encoder. This design enables independent learning of each multi-scale feature.

