# OpenReview forum: "Deep Implicit Optimization for Robust and Flexible Image Registration"
_NeurIPS.cc/2024/Conference — Submitted to NeurIPS 2024_

### Official Review · Reviewer_uXMo · 2024-07-12

**Soundness:** 2
**Presentation:** 2
**Contribution:** 2
**Rating:** 4
**Confidence:** 4

**Summary:**

This paper presents a novel image registration framework that aims to bridge the gap between classical and learning-based approaches. It incorporates fidelity optimization directly into the neural network as a layer. The framework employs end-to-end implicit differentiation through an iterative optimization solver, ensuring that the features learned are both registration and label-aware. Additionally, the warp functions derived are guaranteed to represent local minima of the registration objective within the feature space. The authors report that this framework performs exceptionally well on in-domain datasets and remains robust against domain shifts, such as anisotropy and variations in intensity profiles. Furthermore, the framework is designed to allow seamless switching between different transformation representations at test time without the need for retraining.

**Strengths:**

1. The paper's motivation is both clear and innovative, effectively merging classical optimization techniques with neural networks by embedding an optimization layer within the network. This enhances data consistency and steers the optimization toward local minima.
2. A significant technical achievement of this paper is the backpropagation of gradients through the optimization layer using the implicit function theorem. This demonstrates the paper's technical depth and is the key contribution of the paper.
3. The analysis of loss landscapes provided in the paper is insightful. The flattening of the feature space by neural networks introduces a wider range of possible gradient directions, which, when combined with fidelity loss, enhances overall performance.

**Weaknesses:**

1. A major limitation is the framework's registration accuracy, as measured by the Dice score, which does not demonstrate clear advantages over neural-network-only methods. For instance, in Table 1 on the OASIS dataset, the proposed LKU-Net variant achieves a Dice score similar to that of TransMorph-Large. This calls into question the practical benefit of integrating classical optimization into the network.

2. The absence of specific smoothness measurements in comparison with other methods, such as the percentage of negative Jacobian determinants (||J||<0) or the standard deviation of the logarithm of the Jacobian determinant, is a significant oversight. Without these metrics, it's unclear whether the observed increase in Dice score represents a genuine improvement or merely a trade-off with the deformation field's smoothness.

3. The paper lacks clarity in the reproducibility of results compared to other methods. For example, the learn2reg OASIS leaderboard indicates that LKU-Net can achieve a Dice score of 88.5 on the OASIS dataset without explicitly optimizing for smoothness. Similarly, TransMorph scores 88.5. Additionally, in Table 4, the performance of LKU-Net with Dice supervision is paradoxically worse than without, which is counterintuitive and raises concerns about the implementation fidelity and methodological consistency.

4. While the introduction of an optimization layer with a fidelity function is a key contribution, the paper falls short in comparing its method to other registration methods that also combine learning with optimization. This lack of comparative analysis leaves unanswered questions regarding the true effectiveness and novelty of the proposed method compared to existing approaches.

**Questions:**

1 Could the authors clarify the primary benefits of their framework compared to existing learning frameworks that incorporate optimization, such as ConvexAdam [1], SAMConvex [2], PDD-Net [3], and PRIVATE [4]? Specific examples or quantitative comparisons highlighting how the framework outperforms these methods would be helpful.

2 The paper claims that, "For the first time, our method allows switching between arbitrary transformation representations (from free-form to diffeomorphic) at test time with zero retraining." Could the authors elaborate on why this capability is a significant advantage over other methods? For example, registration networks that separate feature extraction from deformation field estimation using multi-level image pyramids might achieve similar flexibility. What makes the framework distinct or superior in this context?

3 The paper mentions that anatomical landmarks can be incorporated into the framework. Are these landmarks based on image segmentation or keypoints, or do they refer to a more general concept of landmarks? If possible, could the authors demonstrate or provide insights on how these keypoints or landmarks are integrated within the optimization loop?

[1] Siebert, Hanna, Lasse Hansen, and Mattias P. Heinrich. "Fast 3D registration with accurate optimisation and little learning for Learn2Reg 2021." International Conference on Medical Image Computing and Computer-Assisted Intervention. Cham: Springer International Publishing, 2021.

[2] Li, Zi, et al. "Samconvex: Fast discrete optimization for ct registration using self-supervised anatomical embedding and correlation pyramid." International Conference on Medical Image Computing and Computer-Assisted Intervention. Cham: Springer Nature Switzerland, 2023.

[3] Heinrich, Mattias P. "Closing the gap between deep and conventional image registration using probabilistic dense displacement networks." Medical Image Computing and Computer Assisted Intervention–MICCAI 2019: 22nd International Conference, Shenzhen, China, October 13–17, 2019, Proceedings, Part VI 22. Springer International Publishing, 2019.

[4] J. Hu, W. Gan, Z. Sun, H. An, and U. S. Kamilov. A Plug-and-Play Image Registration Network, Mar. 2024. arXiv:2310.04297

**Limitations:**

The paper commendably integrates a fidelity function with an optimization layer into a neural network, but there are several limitations:
1. The optimization layer has not demonstrated significant performance improvements compared to traditional neural network methods on OASIS leaderboard, and the implementation issues in the other datasets/methods raises concerns about the completeness of the result. Clarifying scenarios where this integration is advantageous could enhance the paper's impact.
2. The absence of quantitative smoothness metrics for registration performance is a critical gap. Introducing these metrics would strengthen the comparative analysis with existing methods.
3. The paper lacks a comprehensive comparison with major competitors combines optimization with learning. Expanding this analysis would provide clearer insights into the framework's unique contributions.
4. Concerns about implementation and reproducibility persist.

---

> ### Author Rebuttal · Authors · 2024-08-07
>
> We thank the reviewer for their detailed and insightful feedback. We are excited to hear the reviewer found the paper both clear and innovative, and appreciated the paper’s technical depth, insight provided by Fig.2. We believe we have addressed all concerns, and look forward to engaging in a discussion and convincing the reviewer about the importance of the work.
>
> **A major limitation is the framework's registration accuracy, as measured by the Dice score**
>
> The focus of our paper is not solely achieving the highest Dice score on in-distribution (ID) data. Instead, our primary objective is to develop a robust and flexible method that performs well both ID and under domain shifts such as variations in preprocessing and acquisition configurations. While LKU-Net performs exceptionally well on the OASIS dataset, it completely breaks down under domain shift, which is a significant disadvantage. In contrast, a baseline like SynthMorph (SM) gets a Dice score of ~0.77 on the OASIS validation set, but performs on par with DIO under domain shifts due to its modality-agnostic training methodology. Our approach’s motivation is similar to SM, which aims to maximize performance _and_ robustness, which we believe offers substantial practical benefits in real-world scenarios where data variations are common.
>
>
> **The absence of specific smoothness measurements … trade-off with the deformation field's smoothness.**
>
> These numbers are reported in Table 2. There is a slight (obvious) tradeoff between Dice score and det(Jacobian) values. Table 2 also shows the versatility of our learned feature maps, which do not _overfit_ to the optimization algorithm used during training.
>
> **The paper lacks clarity in the reproducibility ...**
>
> There is a slight difference between the results reported in the papers and the challenge leaderboard, notably due to implementation improvements post-publication. We used the recommended parameters in the official code, and found the numbers to agree (with reasonable error margins) with that of their papers.
>
> **Additionally, in Table 4, the performance of LKU-Net with Dice supervision is paradoxically worse than without**
>
> This is not observed by us alone – [1] show in Sec 4.5.2 that most architectures like Convnet-affine and VTN-affine cannot generalize outside the training domain. Moreover, [1] observe that the semi-supervised models are _inferior_ to their unsupervised models in the LPBA dataset, indicating anatomical knowledge injected to the model with supervision may not generalize well to unseen data beyond the training data. We observe this exact phenomenon with more baselines in Tab.4.
>
> Baselines like SM and LapIRN perform reasonably on OASIS (way worse than LKU), but demonstrate consistent robustness to domain shift. Does it mean they show paradoxical behavior? Very likely not. In general, for registration, the validation overlap on in-distribution data tells nothing about the domain shift performance.
>
> We’re confident that LKUNet is trained properly. We will release all code and models used in the paper for fairness and reproducibility.
>
> **the paper falls short in comparing its method to other registration methods that also combine learning with optimization**
>
> To our knowledge (see L104-L118), most methods do not perform learning and optimization end-to-end. The closest work Reviewer UTrd pointed out is KeyMorph, which cannot be extended to general warp fields which do not have closed-form solutions.
>
> **1 Could the authors clarify the primary benefits of their framework …  methods would be helpful.**
>
> ConvexAdam, SAMConvex, PDDNet use 6D correlation volumes (instead of 4D features) which do not scale well with large data. Moreover, they are not learned end-to-end. PnP uses DEQ to finetune a AWGN network, but the data-fidelity term comes from the intensity images, while our features also incorporate label-fidelity. These are clear differences showing the elegance and strength of our formulation.
>
> **2 The paper claims that, "For the first time, our method allows switching, why is this an advantage over other methods?**
>
> The ability to switch between arbitrary transformation representations (e.g., from free-form to diffeomorphic) at test time without retraining is a significant advantage. Traditional DLIR methods fix the warp field representation during training, limiting flexibility and requiring retraining if the representation needs to change due to different downstream needs (e.g. model was trained with diffeomorphism for normative patients but requires non-diffeomorphism at inference for a subject with pathology). DIO allows for this flexibility, enabling the model to zero-shot adapt to different warp requirements and constraints. This capability is particularly beneficial in scenarios where the desired transformation properties may evolve or vary across applications.
>
> **3 The paper mentions that anatomical landmarks can be incorporated … within the optimization loop?**
>
> Landmark information is incorporated indirectly using a Dice loss (for labelmaps) or landmark distance (for keypoints). Similar to DLIR methods, using a Dice loss at training will learn features such that instance optimization of learned features minimizes dice alignment loss. Therefore, the learned features are label-aware (as seen by the performance gap between ours and ANTs/FireANTs).
>
> At test time, if labels/keypoints are available, instance-optimization can simply add a Dice/landmark distance loss (sparse loss) in addition to feature image alignment loss (dense loss). Hyperparameter tuning can also be done at inference-time.
>
> **Concerns about implementation and reproducibility persist.**
>
> We have provided our entire codebase in the supplementary material. If there are additional concerns, we’re happy to clarify during the discussion period.
>
> [1] Mok, Tony CW, and Albert Chung. "Affine medical image registration with coarse-to-fine vision transformer." CVPR 2022.

---

> > ### Comment · Reviewer_uXMo · 2024-08-14
> >
> > The authors' rebuttal has addressed some concerns, showing the paper's potential for excellence.
> > However, in its current form, it is not yet suitable for publication at NeurIPS.
> > The reviewer encourages the authors to conduct more comprehensive experiments, particularly those involving fidelity loss in the post-optimization phases, rather than simply asserting differences from existing methods.

---

### Official Review · Reviewer_3syP · 2024-07-13

**Soundness:** 2
**Presentation:** 2
**Contribution:** 2
**Rating:** 4
**Confidence:** 4

**Summary:**

This paper introduced DIO, a differentiable implicit optimization layer to a registration network that aimed to bridge the gap of classical-learning-based image registration, considering the incorporation of weak supervision like anatomical landmarks into the learned features. The authors decoupled feature learning and optimization and trained a deep network to predict multi-scale dense features registered through a black box iterative optimization solver.

**Strengths:**

**S1.** The proposed model significantly improves over SOTA models on domain shift experiments.

**S2.** The paper is well-written

**S3.** Multi-scale optimization seems an interesting approach that might be a possible module that can be integrated in deformable image registration.

**Weaknesses:**

**W1. Possibilities of getting artifacts from different voxel sizes.** How do different voxel sizes ensure that the velocity or transformation field is differentiable or invertible? I believe this approach might introduce artifacts and lose fine details when propagating the source image to match the target image. And how processing the image features independently will preserve the diffeomorphism property in generating the transformation field? Can the image matching term efficiently capture the intensity differences likewise treating the input images as pairwise? Overall, treating the input images separately from the feature extractor raises several questions regarding the credibility of the transformation field $\phi^*$.

**W2. Motivation for using multi-scale optimization.** I found the motivation of using multi-scale is somewhat underdeveloped. What is the rationale behind using such kind of optimization considering different source/target image features?


**W3. Applicabilities of learned multi-scale dense features from sparse images.** In Sec. 4 the authors tried to show that DIO learned interpretable dense features and compared to the classical methods DIO preserved the gradient in the loss function. On the other hand, the authors also discussed that deep networks recovered affine transform with $~90$% overlap. I wonder what is the advantage of capturing multi-scale dense features compared to existing DLIR methods such as VoxelMorph, TransMorph, etc.

**W4. Experimental supports.** Though the authors are getting comparable performance in image registration (Tab. 1), they are achieving improved results in testing out of domain/distribution datasets. However, the authors might want to show their model's performance without adapting their proposed multi-scale optimization. Basically, is the optimization scheme or the multi-scale features helping the complete registration model in achieving those bits of improvements? and the important question is why? Two interesting sets of experiments that validate the domain shift hypothesis can be the following -
*(i) train on some of the other datasets (excluding OASIS) and test on the rest,* and
*(ii) train on multiple datasets, including OASIS, and test on the rest.*


Overall, I appreciate the authors for working in the domain of image registration which is very relevant as well as important in the medical imaging domain. However, the current version of the manuscript lacks some important experimental justification and further experiments. With that being said, the current version of the manuscript is under the threshold of acceptance. However, I am open to reconsidering the initial rating if the above concerns are adequately justified.

**Questions:**

How do decoupling feature learning and optimization blocks help the model in getting improved performance over existing DLIR methods?

I tried to summarize all the findings, concerns, and questions in the Weaknesses section.

**Limitations:**

Limitations have been discussed in the paper.

---

> ### Author Rebuttal · Authors · 2024-08-07
>
> We thank the reviewer for their feedback. We are delighted to learn that the reviewer finds the paper well written and appreciates multi-scale feature optimization to be an interesting approach. We believe we have addressed all the remaining concerns and hope the reviewer increases their score and advocates for the acceptance of our paper.
>
> **W1. Possibilities of getting artifacts from different voxel sizes**
>
> This is standard practice in classical image registration, where the fixed and moving images are typically not in the same space. For applications such as unimodal and multimodal registration [1,2,3], ex-vivo to in-vivo registration [4,5], images often differ in size and voxel resolution. Instead of resampling the moving image twice (once to make its size equal to the fixed image and second during warping) thereby introducing more resampling artifacts, classical methods perform a single resampling (during warping only). This is also the motivation for introducing decoupled feature extraction and optimization – to enable registration with modalities with grossly different spatial resolutions.
>
> Many established toolkits like ANTs, NiftyReg, Greedy, Demons, and LDDMM handle varying image sizes effectively. In contrast, current DLIR methods are limited because they require images to have matching spatial dimensions due to channel-wise concatenation. This is a problem with applications like in-vivo to ex-vivo, where downsampling the ex-vivo image loses fine detail and upsampling the in-vivo image consumes memory. Our method overcomes this limitation of existing DLIR methods, allowing for more flexible registration.
>
> **Overall, treating the input images separately from the feature extractor raises several questions regarding the credibility of the transformation field $\phi$**
>
> “this approach might introduce artifacts and lose fine details when propagating the source image to match the target image” – This is indeed the case, but for DLIR methods, which typically resample the images into a uniform spatial resolution (1mm isotropic for OASIS dataset in Learn2Reg) followed by another resampling due to warping – introducing additional resampling artifacts. This loses fine details if the image was acquired at a higher resolution (say 0.7mm in-plane resolution) and introduces resampling artifacts due to anisotropy.
>
> **W2. Motivation for using multi-scale optimization**
>
> Using multi-scale optimization is also standard practice in the classical image registration community. The rationale is that optimization at the finest scale only will converge at some bad local minima, due to the ill-conditioned and ill-posed nature of deformable registration. Therefore, registration at coarser resolutions aligns large structures like ventricles, and finer scales align small structures like gray matter sulci. Motivated by classical methods, certain DLIR methods also use multiscale optimization. Multiscale optimization in DIO is motivated by learning discriminative feature maps at different scales compared to naive downsampled versions of intensity images.
>
> For example, intensity based registration at 4x scale achieves a Dice score of 0.6, whereas our method achieves around 0.75 in the OASIS val set, and enjoys a similar improvement for 2x downsampling. We’ve added this ablation to Appendix due to space issues.
>
> **W3. Applicabilities of learned multi-scale dense features from sparse images.**
>
> We’re not entirely sure what “DIO preserved the gradient in the loss function” and “discussed that deep networks recovered affine transform with 90% overlap” mean. We are happy to clarify in the discussion period.
>
> **W4. Experimental supports.**
>
> We’ve added an ablation on training with and without multiscale optimization. We see slightly better scores for multi-scale optimization, due to better local minima achieved during optimization at coarser scales.
>
> Since our model learns feature maps for registration, it _has to be_ followed by an optimization step. The feature images are indeed helpful - as is evident from the difference between ANTs and our method, wherein the underlying optimizer is the same.
>
> **Two interesting sets of experiments that validate domain shift**
>
> This is exactly what we did, i.e. trained on OASIS and tested on the rest. We did not train on other datasets due to their limited size (364 training images in OASIS versus 40 in LPBA40). Training on multiple datasets is possible, but the size of the OASIS dataset would dominate over the others. Given space considerations, we leave more nuanced ablations for future work.
>
> **How do decoupling feature learning and optimization help the model in getting improved performance over existing DLIR methods?**
>
> Our goal is to not asymptotically outperform existing methods (see Limitations of the paper), but to tradeoff a little bit of in-distribution performance (as in Table1), for generalizable and robust performance under domain shift. Moreover, small improvements in dice score can be attributed to slight misalignments for smaller anatomical regions.
>
> [1] Murphy, Keelin, et al. "Evaluation of registration methods on thoracic CT: the EMPIRE10 challenge." IEEE transactions on medical imaging 30.11 (2011): 1901-1920.
>
> [2] A. Klein, et al. Evaluation of nonlinear deformation algorithms applied to human brain MRI registration. NeuroImage, 46(3):786–802, July 2009.
>
> [3] Wang, Quanxin, et al. "The Allen mouse brain common coordinate framework: a 3D reference atlas." Cell 181.4 (2020): 936-953.
>
> [4] Khandelwal, Pulkit, et al. "Automated deep learning segmentation of high-resolution 7 tesla postmortem MRI for quantitative analysis of structure-pathology correlations in neurodegenerative diseases." Imaging Neuroscience 2 (2024): 1-30.
>
> [5] Meyer, Charles R., et al. "A methodology for registration of a histological slide and in vivo MRI volume based on optimizing mutual information." Molecular imaging 5.1 (2006): 7290-2006.

---

### Official Review · Reviewer_UTrd · 2024-07-17

**Soundness:** 3
**Presentation:** 3
**Contribution:** 2
**Rating:** 4
**Confidence:** 4

**Summary:**

The authors introduce the idea of implicit optimization, and coupled feature extraction for images, to achieve robust image registration. I liked the overall idea and was eager to gain insight into how implicit optimization

**Strengths:**

I really liked the overall ideas here.

An implicit optimization layer does indeed address a DL shortcoming that hs been pointed out before (several DL methods show that some fine-tuning of the output deformation field will improve registration for many domains/moels).

I like the idea of using this in combination with a feature extractor and using those learned features to drive the optimization, with potential benefits down the line.

The paper is also pretty clear and concise, which I like.

There are extensive experiments, with measures of variance, which is a great start.

**Weaknesses:**

Unfortunately, I found the rest of the paper (beyond the core idea) lacking and having several weaknesses.

Importantly, the authors mischaracterize important relevant literature and conceptual ideas, that I think are incorrectly described in the motivation, and not compared to properly in the experiments. This makes it challenging to assess and gain insights from the contribution.

Incorrect characterization: the authors say a few statements that to me seem directly false (and are important to the paper). For example, on line 50 they say "Moreover, design decisions like sparse keypoint learning for affine registration [103, …] do not facilitate dense deformable registration" -- they repeat this in several parts of the paper (e.g. line 107). This is wrong -- their first citation, for example, 103, uses keypoints for *deformable* registration (along with affine). A few lines lower they say "Current DLIR methods are not robust to minor domain shift like varying anisotropy and voxel resolutions, different image acquisition and preprocessing protocols [62, 53, 70, 43]." -- which is incorrect and not supported by the citations. First, citations 62, 53, 70 are from before 2012 and do not discuss DLIR methods at all (but just general registration), whereas citation 43 *is explicitly tackling and achieves robustness to domain shift*. It may well be that the authors' method does better (I am not sure, see below), but the claim is incorrect. Many other papers tackle distribution shift in DL registration -- see Mok et al, 2023. Another crucial omission is related to the authors' claim that DL methods may not output local minima results -- which is true, but plenty of works propose to take the output of neural networks and perform a bit of instance-specific optimization of the resulting field to get to that local minima -- essentially 'fine-tuning' the field for a couple of seconds (e.g. VoxeMorph TMI 2019, but plenty of other methods as well after that for example from Matthias Heinrich's group). This is crucial to the current paper, since the proposed method essentially does the same thing at inference -- runs a forward neural network (albeit just as a feature extractor), and then performs an optimization for the image pair -- and while it's done differently (and more elegant in some sense) than the existing literature, these approaches are very related. Overall, I was excited about the method but overall found the motivation/related-works either misleading or lacking rigor -- perhaps the authors are simply not aware of the abilities of existing literature mentioned above, but this does  limit the novelty and insight substantially

In the experiments, it seems to me that some obvious results are missing:
. I am not sure why methods used in Figure 3 (e.g. [43]) are missing from Table 1. It seems like a crucial comparison. Deformable KeyMorph [103] is missing from the whole experiments section, and is close to the existing method in that it separates the feature extraction (there via keypoints, but using a parallel net) from the optimization. Training keymorph on oasis and testing it seems like an important comparison if we are to extract some insights into how to decisions in the proposed method (the feature extractor and the implicit optimization) improve our insights in the field.
. Overall the results in Table 1 do not seem impressive -- comparable at best with existing methods. This is totally okay in my book, if the authors are able to communicate other interesting insights. Unfortunately, I do not believe this is the case.
. In the domain-shift section the authors show that their method tens to outperform the DL methods. However, their method gets the benefit of doing instance-specific optimization (the proposed layer) after feature extraction, at the cost of some GPU work for each pair. This is what instance-specific optimization does at the end of DL methods (as discussed below), which was employed in several papers, but this is not included in the comparison! This is a peculiar omission to me -- it should be included for completeness, but importantly it is also crucial to understand whether the proposed method behaves differently -- perhaps there is some advantage, in several situations, to the proposed method, or perhaps it offers more guarantees, etc -- we simply don't know. Minor: it would also be interesting to understand what are the limits of this domain shift of this model -- does it generalize to more substantial variations in modality, or 7mm slice spacing found in clinical sequences?
. I also find the claim that DL methods do not work without crops peculiar - most DL methods are convolutional and hence not size specific, and some (e.g. SynthMorph, which the authors refer to here) does not even require both images be of the same size.
. Since one of the contributions of the paper is the parallel feature extraction (to be used with the optimizer), it seems to me that it would be an important ablation to take the features of some robust method (does not have to be registration, even a domain-shift-robust segmentation network will do, or a 'robust foundation model', etc) and see if that can be combined with an optimizer. This would help provide insights if the formulation of the proposed model and the end-to-end training is useful.
. The claim in 4.4 is also missing some reaosnable comparison -- would it not be possible to take the displacement field of any other DL method, initialize your favorite parametrization (freeform, diffeomorphic, etc) with that, and run the (any) optimizer? It seems like this is easily doable and a reasonable comparison?

**Questions:**

Please see comments above. It would be important to understand why the omissions in the motivation and experiments happened -- did the authors not know some of the keypoint based methods are deformable? Were they not aware of instance-specific optimization after the output of a DL method (which is discussed in several papers)? etc.

**Limitations:**

Yes the paper has a limitations section. It would be nice if the authors could comment on how this method can be used on CPUs -- by far the standard hardware available to non-ML users (neuroscientists, clinicians, etc).

---

> ### Author Rebuttal · Authors · 2024-08-07
>
> We thank the reviewer for their insightful feedback, and we’re glad to note that they found the paper overall very interesting. The review has been immensely helpful in improving the quality of our work. We believe we have addressed all concerns in the rebuttal, and we look forward to a discussion and convincing the reviewer about the importance and relevance of our work.
>
> **Sparse keypoints (kps) don’t facilitate dense reg. is wrong - KeyMorph (KM) does deformable reg. with sparse kps**
>
> This is an oversight on our part. We have revised this incorrect characterization in the paper, and mentioned that these formulations allow linear and deformable registration. However, our motivation for DIO still holds, i.e. KM can only output deformation fields that admit closed-form warps (e.g using TPS), from learned keypoints. However, TPS is a limited class of warps, cannot be guaranteed to be diffeomorphic, and a vast majority of widely used parameterizations (free-form, SVF, geodesic, LDDMM, SyN) do not admit closed form solutions making KM inapplicable. While in KM the keypoints are trained end-to-end using backprop, we use implicit differentiation through the optimization to learn dense feature maps. DIO is therefore an elegant and generalized extension of the KM idea: “keypoints” are replaced by “dense feature maps” and “closed-form warp field” parameterizations are replaced by “arbitrary warp fields” (i.e. solutions of optimization objectives). Moreover, KM runs out of memory quickly – TPS with 512 kps runs OOM on a A6000 GPU whereas DIO outputs multi-scale feature maps equivalent to 192 * 224 * 160 * (1 + 1/8 + 1/64) * 16 / 3 ~ 41M keypoints.
>
> **acquisition and preprocessing protocols [62, 53, 70, 43]…not supported by the citations**
>
> We agree! The first three citations were supposed to reference different image acquisition and preprocessing protocols themselves (neuro/lung imaging). However, the statement itself is correct as shown in Fig.3. and Tab.4; these are very strong results to support the statement. We have cited Fig.3 in the paper instead.
>
> **DL methods dont output local minima, but instance optimization (IO) can – this is very related**
>
> We agree that these approaches are related, however there is a subtle but crucial difference. In existing approaches, the deep network is “label-aware”, but outputs a warp field that may not be a local minima of *any* objective. IO is performed on the *intensity images* that are not label aware. This IO is a small change on top of the predicted warp field. In contrast, we train feature maps to be label aware *by design* (since registering feature maps minimize label overlap loss), as shown in Tab.1 and Fig.3. We sidestep warp field prediction altogether and perform a single IO step on *label aware features* instead of *label-unaware intensity images*.
>
> **Synthmorph is missing in Tab1**
>
> For DLIR, we only added a few recent methods that score > 0.82 Dice on OASIS val. SM scores a ~0.77 Dice on the OASIS dataset, with the provided pretrained model.
>
> **Deformable KeyMorph is missing from the whole experiments section**
>
> We were aware only of the affine variant. However, the performance of KM is unsatisfactory. We trained KM on the OASIS train split, and observed the following dice scores for OASIS val and zero-shot on IBSR:
>
> | Loss | OASIS | IBSR |
> |--|--|--|
> | MSE | 0.6078| 0.4663 |
> |Dice| **0.6437** | 0.49250 |
>
> The in-distribution performance strongly agrees with deformable registration dice scores in Fig.5,7,8,9 in the KM paper. We observe that KM is very robust in recovering from large rotational deviations due to its training scheme (good for canonicalizing data acquired on non-standard coordinates) , but is an unsatisfactory baseline for deformable registration due to ~6M voxel displacements being parameterized by only 512 keypoints - making registration of small subcortical structures especially hard.
>
> **In the domain-shift section the authors show that their method works because of IO, but others dont use IO**
>
>
> This experimental design is justified by the ‘feature space’ in which IO is performed. We find the warp field as the minimizer to an objective function with feature maps. The minimizer is found using IO. However, the warp field produced is a local minimizer of IO in the “learned feature space” and not in the “intensity space”. The warp field produced by DIO may not be a minimizer in the intensity space. Moreover, implementing IO for every baseline is a highly tedious task, since different baselines have different implementations of the warp field, including scaling, different coordinate systems, etc.
>
> A fair comparison for DIO would be to perform IO of learned features, followed by IO of intensity images. For simplicity, we do not perform IO in the intensity space for any method.
>
> **some (e.g. SM, which the authors refer to here) does not even require both images be of the same size.**
>
> This is not the case, since every DLIR method requires the fixed and moving images to be concatenated channelwise, implying matching spatial dimensions. IBSR18 volumes have different voxel sizes. Moreover, some of the methods have hardcoded image sizes – this is an implementation issue. This also highlights challenges in using DLIR methods from official implementations. Methods like TransMorph use ViT end-to-end and cannot be used with different image sizes. We will make all scripts public for fairness and reproducibility.
>
> **it seems to me that it would be an important ablation to take the features of some robust method … end-to-end training is useful.**
>
> This seems like a useful ablation, but we are not sure which method to use for feature extraction.
>
> **The claim in 4.4 is also missing some reasonable comparison**
>
> We’re not sure what that would show. Sec. 4.4 is written to show that the learned features in DIO do not “overfit” to the warp field representation during training, shown by inference-time switching to an unseen optimizer.

---

> > ### Comment · Reviewer_UTrd · 2024-08-13
> >
> > I read the rebuttal carefully and thank the authors.
> >
> > However, I believe the various omissions require a new submission, since they would require a new revision. They are simply too central to the work, and cannot be explained in a single rebuttal paragraph -- in a normal journal submission, this would be a major revision.  For example, the KM omission requires a bit more than a simple run through and a 2x2 table -- this assumption that there is no related method is repeated throughout the paper and used to justify decisions. The KM method would need to be more thoroughly tested. Similarly, while IO is indeed not exactly the current method, the entire paper ignores IO as if it doesn't exist -- when in fact the difference is subtle at best. This not only requires reframing, but rigorous experimentation. This line, for example, i think substantially weakens the paper and I would argue is an important omission, still: "For simplicity, we do not perform IO in the intensity space for any method."
> >
> > For a future submission, some more minor comments:
> > - the citations example was more minor, but please note that there are several statements back to back like this -- where the claim is false and the citations used (that make it look like you are using related works to back up a claim) do not support the claim. I don't understand how the authors stand by the truthfulness of the (in my opinion over-reaching) statement -- there are definitely DLIR methods that are robust to domain shift, or at least some types of domain shift. In my opinion, Figure 3 does *not* support the statement -- yes the current method might improve on some methods, but only marginally -- making the claim way too strong.
> > - but please note that while it's true that most DLIR methods concatenate the inputs in the architecture, some available implementations take in different-sized images, resample to the same (highest) voxel size to be isotropic, pad, and only then concatenate.  I believe SM does this, but am not an expert. Either way, it's trivial to do. This enables the method to work with differently-sized inputs. I am not using this point in my decision, though -- just a note for your submission.
> >
> > Unfortunately, overall I believe the core omissions, which still exist (e.g. see the IO comment), are central to the work and would require a new submission to bre properly reviewed. I believe my original score stands. I do wish the authors good luck with this, I think there is something good here to be communicated, just not in its current form.

---

### Author Rebuttal · Authors · 2024-08-07

We thank all reviewers for their insightful feedback and for taking time to improve the quality of our work. We are glad that reviewers found the overall idea [UTrd,  ] and multi-scale optimization idea [3syP] interesting, innovative and insightful [uXMo], clear and concise writing [UTrd, 3syP, uXMo] , extensive experiments with measures of variance [UTrd], improvement over SOTA models in domain shift [3syP], demonstrating technical depth [uXMo].  We have addressed all questions in the individual comments. We summarize and clarify some common concerns:

**Performance in Table 1 is not outperforming LKU.**

Our goal is to not outperform existing methods (see Limitations of the paper), but to trade a little bit of asymptotic performance on the in-distribution dataset (as in Table1), for accurate, generalizable and robust performance under minor domain shift, interpretability, and zero-shot plug-and-play of arbitrary displacement field constraints and optimizers. Our model still performs very competitively on in-distribution dataset.

In real-world settings, registration algorithms must be robust to variations in data distribution. Effective registration algorithms should be robust and interpretable, allowing us to understand why registration might fail if it does. We aim to extend these capabilities in DLIR methods by learning feature maps that lead to robust, accurate registration and interpretable features. This practical benefit is significant compared to methods that perform well on a single test data distribution but fail under real-world variations.

Given the unanimous agreement on the technical novelty, depth, and innovative ideas in the paper, accompanied by clear and concise writing, we believe the paper’s merit should not be determined solely by asymptotic in-distribution validation performance but by its overall applicability to real-world registration scenarios determined by performance under domain shift, interpretability, and flexibility of the algorithm to varying or evolving registration scenarios.

**Comparison with related baselines**

To our knowledge (see L104-L118), most existing methods do not perform learning and optimization end-to-end. The closest work Reviewer UTrd pointed out is KeyMorph, which uses closed-form warp functions (i.e. thin plate splines) and cannot be extended to general displacement fields that do not have closed-form solutions.
We add comparisons with KeyMorph (KM) both in-distribution on the OASIS validation set, and on domain shift.

| Training Loss | OASIS | IBSR |
|--|--|--|
| MSE | 0.6078| 0.4663 |
|Dice| **0.6437** | 0.49250 |

The in-distribution performance strongly agrees with deformable registration _dice scores_ in Fig.5,7,8,9  in the KM paper. We observe that KM is very robust in recovering from large rotational deviations due to its training scheme (good for canonicalizing data acquired on non-standard coordinates as shown in their paper) , but is an unsatisfactory baseline for deformable registration due to (192x160x224)~6M voxel displacements being parameterized by only 512 keypoints - making registration of smaller subcortical structures especially hard. This is the maximum number of keypoints used in the KeyMorph paper as well; adding more keypoints runs out of memory on an A6000 GPU with 48GB VRAM – indicating steep memory requirements.

We use the official repository (https://github.com/alanqrwang/keymorph) for training KeyMorph.

For training _without_ label supervision, we use the following script:
```
python scripts/run.py --job_name oasis_unsup --save_dir ./oasis-unsup --num_keypoints 512 --loss_fn mse --transform_type tps_0 --data_path ./oasis_data.csv --train_dataset csv --run_mode train --backbone truncatedunet --use_amp
```

For training _with_ label supervision, we use the following script:
```
python scripts/run.py --job_name oasis_sup --save_dir ./oasis-sup --num_keypoints 512 --loss_fn dice --transform_type tps_0 --data_path ./oasis_data.csv --train_dataset csv --run_mode train --backbone truncatedunet --use_amp
```
Each training script runs for 2000 epochs (default), which takes about a day.


In regards to writing, we have addressed all changes in individual comments.

---

### Decision · Program_Chairs · 2024-09-25

**Decision:**

Reject

**Comment:**

The paper advocates a novel differentiable implicit optimization layer for image registration networks. While all reviewers agree that this is a quite interesting idea, major issues remain even after the rebuttal and discussion phase. I agree with the overall consensus that adjusting the current manuscript to reflect the required changes (e.g., inclusion of omissions) would require a major revision, which cannot be guaranteed within the NeurIPS review cycle. For that reason, I am recommending rejection at this point, but I also would like to encourage the authors to continue this work and revise accordingly for a resubmission.